# Striking Differences in Platelet Distribution between Advanced-Platelet-Rich Fibrin and Concentrated Growth Factors: Effects of Silica-Containing Plastic Tubes

**DOI:** 10.3390/jfb10030043

**Published:** 2019-09-17

**Authors:** Tetsuhiro Tsujino, Hideo Masuki, Masayuki Nakamura, Kazushige Isobe, Hideo Kawabata, Hachidai Aizawa, Taisuke Watanabe, Yutaka Kitamura, Hajime Okudera, Kazuhiro Okuda, Koh Nakata, Tomoyuki Kawase

**Affiliations:** 1Tokyo Plastic Dental Society, Kita-ku, Tokyo 114-0002, Japan; tetsudds@gmail.com (T.T.); hideomasuki@elm-dc.com (H.M.); maoh4618@me.com (M.N.); kaz-iso@tc4.so-net.ne.jp (K.I.); hidei@eos.ocn.ne.jp (H.K.); sarusaru@mx6.mesh.ne.jp (H.A.); watatai@mui.biglobe.ne.jp (T.W.); shinshu-osic@mbn.nifty.com (Y.K.); okudera@carrot.ocn.ne.jp (H.O.); 2Division of Periodontology, Institute of Medicine and Dentistry, Niigata University, Niigata 951-8514, Japan; okuda@dent.niigata-u.ac.jp; 3Bioscience Medical Research Center, Niigata University Medical and Dental Hospital, Niigata 951-8520, Japan; radical@med.niigata-u.ac.jp; 4Division of Oral Bioengineering, Institute of Medicine and Dentistry, Niigata University, Niigata 951-8514, Japan

**Keywords:** advanced platelet-rich fibrin, concentrated growth factors, platelets, CD41, centrifugal force, silica

## Abstract

Compared with platelet-rich plasma, the preparation of platelet-rich fibrin (PRF) is simple and has not been overly modified. However, it was recently demonstrated that centrifugation conditions influence the composition of PRF and that silica microparticles from silica-coated plastic tubes can enter the PRF matrix. These factors may also modify platelet distribution. To examine these possibilities, we prepared PRF matrices using various types of blood-collection tubes (plain glass tubes and silica-containing plastic tubes) and different centrifugation speeds. The protocols of concentrated growth factors and advanced-PRF represented high- and low-speed centrifugation, respectively. Platelet distribution in the PRF matrix was examined immunohistochemically. Using low-speed centrifugation, platelets were distributed homogeneously within the PRF matrix regardless of tube types. In high-speed centrifugation, platelets were distributed mainly on one surface region of the PRF matrix in glass tubes, whereas in silica-coated tubes, platelet distribution was commonly more diffusive than in glass tubes. Therefore, both blood-collection tube types and centrifugal conditions appeared to influence platelet distribution in the PRF matrix. Platelets distributed in the deep regions of the PRF matrix may contribute to better growth factor retention and release. However, clinicians should be careful in using silica-coated tubes because their silica microparticles may be a health hazard.

## 1. Introduction

Platelet-rich fibrin (PRF) has been applied increasingly in regenerative dentistry and orthopedic surgery. PRF is derived from platelet-rich plasma and is characterized by its simple preparation protocol that does not employ anticoagulants or coagulation factors [1,2,3,4,5,6,7]. However, the reproducibility and predictability of PRF therapy remains controversial, especially in the context of bone regeneration [8]. Therefore, regulatory authorities in many countries still stipulate the need for high-quality evidence obtained from randomized clinical trials and meta-analyses.

Two major approaches have been proposed to improve this situation: Standardization of the preparation protocol and quality inspection of individual PRF preparations. Ultimately, quality control has to depend on the latter approach. To aid in the realization of individual quality inspection, we have developed a method to directly determine platelet counts in an insoluble PRF matrix [9]. However, it is still difficult to evaluate individual PRF matrices in a timely manner in the clinical setting. In contrast, effective standardization of preparation and practical application does not assure the quality of individual PRF matrices but is expected to efficiently minimize variability and maximize efficacy in the same blood samples, consistent with other types of platelet concentration protocols [10,11,12,13,14]. Thus, we previously proposed the necessity of standardization in terms of both preparation and clinical application [15]. 

To establish the effective standardization of preparation protocols, the resulting PRF matrices need to be characterized in detail in each protocol. Because the PRF preparation protocol is quite simple, major modifications can be made only in the centrifugation conditions (i.e., speed, time, and rotor types) and tube types. To date, several major manual PRF preparation protocols, such as the protocols for the original Chourkroun’s PRF, which is also designated as leukocyte-rich PRF), advanced-PRF (A-PRF), concentrated growth factors (CGFs), and injectable-PRF, have been developed mainly by modifying the speed of centrifugation [16]. PRF matrices prepared by these protocols are often compared based on fibrin fiber morphology, mechanical strength, degradation, and growth factor release [17,18,19]. In a previous study [18], we found no significant difference in mechanical strength, structure of fibrin mesh, or degradation properties between A-PRF and CGF preparation protocols. On the other hand, Kobayashi et al. [19] and Dohan Ehrenfest et al. [17] have reported conflicting data regarding growth factor release. Thus, it remains to be clarified which type of PRF matrix demonstrates superior growth factor-carrying properties; it is plausible that centrifugal speeds influence growth factor retention and release. 

A fundamental problem has necessitated modification of PRF preparation protocols. Inconveniently, the recent shortage of plain glass blood-collection tubes has required clinicians to use alternative silica-containing plastic tubes. This tube shift is based on the assumption that PRF prepared by such plastic tubes are identical to that made by glass tubes mainly in terms of growth factor retention and release. However, even though silica microparticles are thought to function to coagulate blood as a glass surface does, no one has clearly shown this. In a previous study [20], we demonstrated that silica microparticles are detached from inner walls of tubes or films contained in tubes during blood collection and centrifugation, and are subsequently incorporated into the resulting PRF matrix. Therefore, regarding contamination of silica microparticles, the PRF matrices are not identical. In addition, it is possible that the released silica microparticles may significantly modify platelet distribution in PRF matrices prepared by those plastic tubes.

This study was performed to demonstrate possible differences in platelet distribution in PRF matrices prepared by different protocols and using different tube types. We believe that the manner of platelet distribution is one of major factors that regulates growth factor retention and release in the PRF matrix. If so, the present data will clarify of mechanisms of their growth factor retention and release, and will inform improvements of PRF therapy.

## 2. Materials and Methods 

### 2.1. Preparation of PRF Matrices

After obtaining individual written informed consent, blood samples were collected, without anticoagulants, from 10 non-smoking healthy male volunteers aged 30 to 63 years. The study design and consent forms for all procedures (project identification code: 2297) were approved by the ethics committee for human participants at the Niigata University School of Medicine (Niigata, Japan) on 14 October 2015, in accordance with the Helsinki Declaration of 1964, as revised in 2013.

Fresh blood samples (~9.0 mL) from each donor were collected into two varieties of vacuum plain glass tubes, A-PRF+ (Jiangxi Fenglin Medical Technology Co. Ltd., Fengcheng, China) and Becton-Dickinson (BD) Vacutainer (Product code: PRF366430; Becton, Dickinson and Company, Franklin Lakes, NJ, USA), plastic tubes containing a silica-coated film (Venoject II VP-P100K; Terumo, Tokyo, Japan), and silica-coated plastic tubes (Neotube NP-PS0909, Nipro, Osaka, Japan). Blood was immediately centrifuged at 200× *g* for 14 min (A-PRF protocol) using a Duo centrifuge (Process for PRF, Nice, France) or by the CGF protocol using a program that automatically changes the centrifugal speed as follows: 30 s, acceleration; 2 min, 692× *g*; 4 min, 547× *g*; 4 min, 592× *g*; 3 min, 855× *g*; 36 s, deceleration. This CGF protocol was carried out using a Medifuge centrifugation system (Silfradent S. r. l., Santa Sofia, Italy). All centrifugation was performed at ambient temperature (22–25 °C) and all centrifugal conditions are summarized in Table 1.

Quality checks were carried out on individual blood samples by performing platelet and other blood cell counts using a pocH 100iV automated hematology analyzer (Sysmex, Kobe, Japan).

### 2.2. Immunohistochemical Examination

Freshly prepared PRF clots were gently, but not fully, compressed with a stainless-steel PRF compression device (PRF stamper; JMR Corp. Ltd., Niigata, Japan) [15], washed three times with Phosphate Buffered Saline (PBS), and fixed in 10% neutralized formalin. After being divided into 7 pieces (Figure 1a: A-PRF), the fixed PRF membranes were dehydrated in a series of ethanol washes, embedded in paraffin, and sectioned at a thickness of 6 μm.

Localization of platelets in PRF matrices was determined using a previously described immunohistochemical method [15], outlined here: Deparaffinized sections were antigen-retrieved using Liberate Antibody Binding Solution (Polysciences Inc., Warrington, PA, USA) for 15 min and blocked with 0.1% Block Ace (Sumitomo Dainippon Pharma Co., Ltd., Osaka, Japan) in 0.1% Tween-20-containing PBS (T-PBS) for 1 h. The specimens were then probed with a rabbit polyclonal anti-CD41antibody (GeneTex, Irvine, CA, USA), diluted 1:400 in ImmunoShot Mild (CosmoBio Co., Ltd., Tokyo, Japan), overnight at 4 °C. This was followed by incubation with horseradish peroxidase-conjugated goat anti-rabbit IgG antibody (Cell Signaling Technology, Danvers, MA, USA) (1:100 diluted in T-PBS) for 1 h at ambient temperature. Immunoreactive proteins were visualized following the addition of 3,3’-diaminobenzidine (DAB) substrate solution (Kirkegaard & Perry Laboratories, Inc., Gaithersburg, MD, USA).

A second section from each set of conditions was stained with hematoxylin and eosin (HE) to observe the microstructure of each PRF matrix.

## 3. Results

Figure 1b shows the photomicrographs of A-PRF cross-sections at lower magnifications. Individual sections, except for both ends, were subjected to further immunohistochemical examination.

Appendix A shows the platelet distribution in the PRF matrix prepared using glass (A-PRF+) tubes by low- (a: A-PRF protocol) and high-speed centrifugation (b: CGF protocol). The upper margins, to which blood cells and serum proteins were attached, represent the region facing the inner wall of tubes. Following the A-PRF protocol (low-speed centrifugation), CD41^+^ platelets were distributed diffusely over all regions of the PRF matrix (Appendix A). Although only regions 2, 4, and 6 are shown in the figure, these are representative of platelet distribution in all regions. In contrast, in samples prepared using the CGF protocol (high-speed centrifugation), CD41^+^ platelets were distributed mainly around the upper peripheral region in the figure. Other CD41^+^ platelets were distributed sparsely in the deep region and around the lower peripheral region (Appendix A).

Appendix A shows the platelet distribution in the PRF matrices prepared using glass (BD Vacutainer) tubes by low- (Appendix A: A-PRF protocol) and high-speed centrifugation (Appendix A: CGF protocol). As with A-PRF+ tubes, low-speed centrifugation distributed CD41^+^ platelets diffusely throughout the PRF matrix, whereas high-speed centrifugation lead to localized distribution on the side receiving gravity force and in closer proximity to the red blood cell fraction.

Appendix A shows the platelet distribution in the PRF matrix prepared using plastic tubes containing silica-coated film (Terumo Venoject II), by low- (Appendix A: A-PRF protocol) and high-speed centrifugation (Appendix A: CGF protocol). Again, low-speed centrifugation resulted in diffuse distribution of CD41^+^ platelets throughout the PRF matrix. However, high-speed centrifugation did not clearly localize CD41^+^ platelets around the upper side. CD41^+^ platelets were, to some extent, distributed in the deep regions in this PRF matrix. Such platelet distributions were distinguishable from those of the PRF matrix types described above.

Appendix A shows the platelet distribution in a PRF matrix prepared using silica-coated plastic tubes (Neotube) by low- (Appendix A: A-PRF protocol) and high-speed centrifugation (Appendix A: CGF protocol). As with the other tube types, low-speed centrifugation distributed CD41^+^ platelets diffusely throughout the PRF matrix. High-speed centrifugation also distributed CD41^+^ platelets well homogenously in both peripheral and deep regions.

These findings are summarized in Figure 2.

## 4. Discussion

The present data reveal two important points. First, the striking differences in platelet distribution between the silica-coated plastic tubes and conventional plain glass tubes, especially at high-speed centrifugation. Second is the predicted difference in platelet distribution between the A-PRF and CGF protocols, of which the major difference is the centrifugation speed.

### 4.1. Practical Difficulty of PRF Preparation

It is recommended that only approved devices and centrifuges should be used to prepare a specific type of PRF matrix. For example, leukocyte-rich PRF (L-PRF) preparation protocols originally indicated the use of the centrifuge and “glass tubes” provided by Intra-Spin (Intra-Lock International Inc., Boca Raton, FL, USA). L-PRF prepared using the same centrifugal conditions with an alternative brand of glass tubes is not considered genuine in the industry. This is largely a quality-control measure to avoid the impact of poor-quality tubes and inaccurate or unstable centrifugation on the end product. For example, when the inner walls of glass tubes are over-siliconized, coagulation is suppressed or delayed due to blockade of the contact between the glass surface and coagulation factor XII [21]. However, ironically, “plastic tubes” with silica-coating are now “officially” provided as alternative glass tubes to clinicians for preparation of L-PRF (for possible health hazardous effects of silica, see Section 4.3). To our knowledge, at present, only the Process for PRF provides plain glass tubes for preparation of PRF, especially for its brand A-PRF preparation, or other types of platelet concentrates for regenerative therapy.

Here, we demonstrated that regardless of the brands of the glass tubes used, the same centrifugal conditions result in a comparable distribution of platelets in the CGF and A-PRF matrices prepared by the high- and low-speed centrifugation protocols, respectively. Although we did not evaluate all commercially available glass tubes, we think that these findings are representative of all plain glass tubes approved by regulatory authorities, such as United States Food and Drug Administration, European Medicines Agency, and Japan Ministry of Health, Labor, and Welfare. As we sometimes experienced failure of clot formation in “non-approved, evacuated plain glass tubes for blood collection” that were imported and sold “for use of chemical experiment only”, we suggest that clinicians should not use non-approved glass tubes without quality testing.

### 4.2. Differences in Platelet Distribution by Tube Types

The decision by major international medical device manufacturers to stop producing glass blood-collection tubes has caused practical difficulties associated with PRF preparation in clinical settings. Assuming that silica-coated plastic tubes can be used as a reliable alternative of plain glass tubes, many clinicians actually use silica-coated plastic tubes for PRF preparation. However, the present data support our previous demonstration that their assumption is wrong [20]. Silica-coated tubes can distribute platelets relatively widely in the PRF matrix regardless of centrifugal speeds, whereas glass tubes distribute platelets depending on centrifugal speeds (see Section 4.4.). This difference is probably due to release of silica microparticles and ubiquitous initiation of coagulation and platelet activation.

### 4.3. Concern with Possible Risks of Silica-Coated Tubes

Though the Nipro brand of tubes (Neotube) evaluated in this study are marketed for “in vitro diagnostic use only”, we consider them to be as clinically relevant as tubes manufactured by BD and Terumo following the publication of research papers reporting the use of similar silica-coated tubes for PRF preparation [22,23,24]. 

In this study, we observed that silica micro- or nanoparticles were easily detached from the inner wall of the tube during blood collection [20] and during retrieval of the PRF clot. Silica microparticles detached from the inner wall of plastic tubes are thought to cause ubiquitous activation of both the intrinsic coagulation pathway and platelets, a possibility which is supported by the finding that platelets were distributed relatively more widely even after high-speed centrifugation.

This phenomenon also highlights the potential risks of silica particle contamination in the PRF matrix, which we very recently demonstrated [20], and at the site of implantation. Although they have been regarded as chemically stable, silica nanoparticles were recently reported to induce cytotoxicity and inflammatory responses in lung epithelial and endothelial cell lines and in hippocampal cells [25,26,27]. The proposed mechanisms involve interactions between silica nanoparticles and the cell surface and the generation of reactive oxygen species [25]. A recent report described that specific types of amorphous silica nanoparticles can act as tumor-promoting substances [28]. As such toxic effects are thought to depend on the molecular structure (i.e., amorphous or crystalline), particle size, concentration, and surface charge of silica, it is difficult to predict how silica microparticles that detach from tubes will influence the surrounding tissue and cells at the implantation sites. However, because in a parallel study [Kawase et al., manuscript in submission], we found that silica microparticles obtained from silica-coated plastic tubes have substantial cytotoxic effects, we recommend that clinicians observe the manufacturers’ caution of “in vitro diagnostic use only” and not use silica-coated tubes for PRF preparation.

### 4.4. Differences in Platelet Distribution by Centrifugal Speeds and Tube Types

In general, as the centrifugal force increases, blood cells are more sharply fractionated. Therefore, in this study, low-speed centrifugation was expected to induce diffusible distribution of platelets throughout the PRF matrix. However, the impact of the preceding fibrin fiber formation on platelet distribution remains unknown. We demonstrated that, as expected, low-speed centrifugation distributed platelets almost uniformly in the resulting PRF matrix. In contrast, both silica-containing tubes distributed platelets more homogenously in low-speed centrifugation than plain glass tubes. Also, in high-speed centrifugation, the Nipro silica-coated tubes, but less so with the Terumo tubes containing silica-coated disk, distributed platelets more homogenously than predicted. This difference is probably due to the difference in the amounts of silica microparticles; the amount of silica microparticles in Nipro Neotubes is substantially greater than that in Terumo Venoject II [20].

These findings raise further questions regarding growth factor distribution. While the nature of growth factor retention in PRF matrices is poorly understood, some theories have been proposed. The general consensus is that significant levels of growth factors are present in the exudate in a liquid form, while the remainder are contained within platelets and/or adsorbed to fibrin fibers. It is plausible that this minor portion of growth factors within or surrounding platelets are present in an insoluble form, which may potentially contribute to the delayed release of growth factors from PRF matrices. To date, there have been two conflicting articles published comparing A-PRF and L-PRF [17,19]. Kobayashi et al. demonstrated that A-PRF is superior to L-PRF in terms of retention capacity and release capability of growth factors [19], whereas Dohan Ehrenfest et al. reported that A-PRF has a very low capacity as a growth factor carrier [17]. Due to a lack of convincing evidence, we sought to address this disparity in our study.

Our data show that platelets are entrapped and enriched more efficiently in the A-PRF matrix than in the CGF matrix, supporting the findings of the former authors. However, Dohan Ehrenfest et al. also suggested that high centrifugal force activates platelets to release their growth factors through activation of leukocytes [17]. In a subsequent study, this group proposed that the key issue in platelet concentration technologies is not the quantity of platelets, but the interlinkage between platelet, leukocytes, fibrin, and growth factors [29]. Therefore, a more careful investigation is required to provide clear evidence of growth factor distribution to cease such “evidence-unbased” debate.

## 5. Conclusions

Due to similarities in growth factor contents and mechanical strength [18,30], we had previously concluded that A-PRF matrix prepared by low-speed centrifugation and CGF matrix prepared by high-speed centrifugation using glass tubes would be almost identical. However, the present data suggest that tube types and centrifugal speeds substantially influence platelet distribution in their resulting PRF matrices. Even if this difference does not influence the initial growth factor contents, it may influence the nature of growth factor retention and release. In addition, regardless of the possible superiority, silica-coated tubes should not be used to avoid possible health hazards in clinical settings.

## Figures and Tables

**Figure 1 jfb-10-00043-f001:**
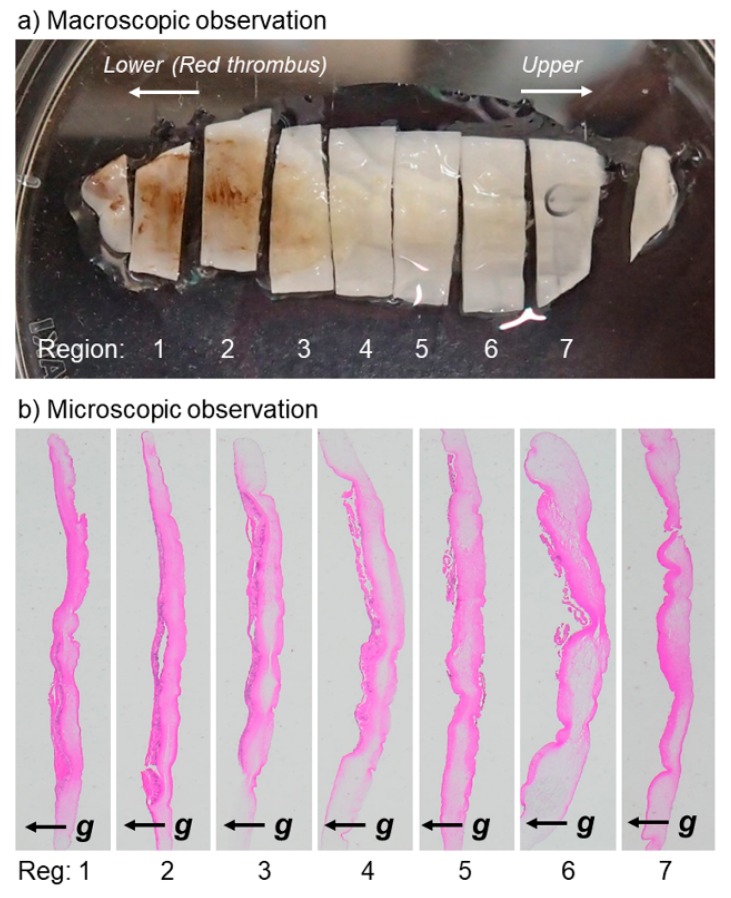
Macroscopic observation of a compressed and fixed A-PRF membrane. (**a**) This PRF membrane was divided into seven pieces, designated as region 1 to 7, where region 1 represents the region closest to the red blood cell fraction. (**b**) Microscopic observation of A-PRF cross-sections obtained from individual regions. Cross-sections were stained with Hematoxylin and eosin (HE). To confirm morphological similarity, the magnitude of sections was modified to adjust their lengths at similar levels. Arrows represent the direction of gravity force.

**Figure 2 jfb-10-00043-f002:**
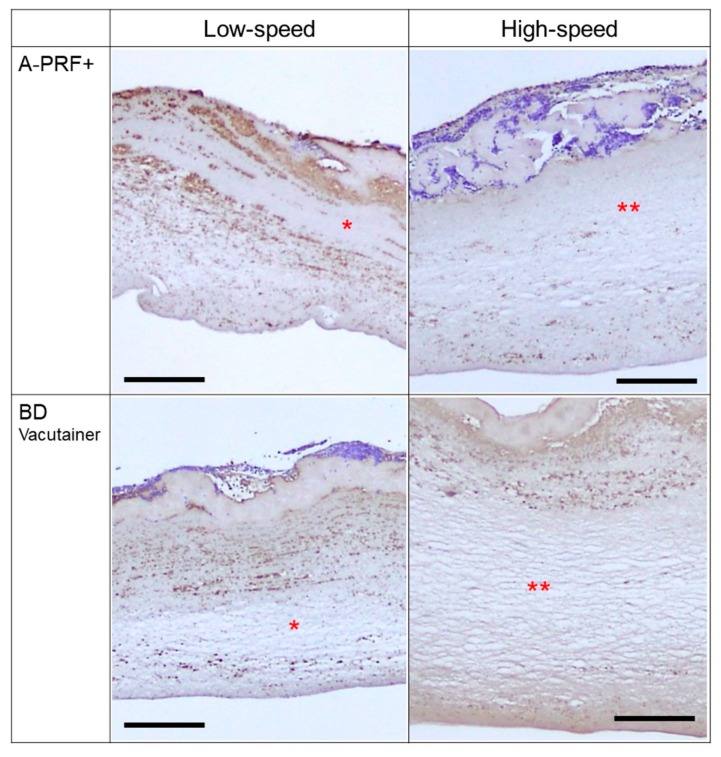
Summary of platelet distribution under various conditions. Asterisks represent wide-open spaces, where platelets were distributed sparsely. Double asterisks represent wider spaces than single asterisks. Bar scales are 200 µm.

**Table 1 jfb-10-00043-t001:** Centrifugal conditions and the corresponding data.

Tube Types (Manufacturer)\Centrifugation	Low-Speed (A-PRF Protocol)	High-Speed (CGF Protocol)
Plain glass tube (A-PRF+)	Appendix A ^1^	Appendix A
Plain glass tube (BD Vacutainer)	Appendix A	Appendix A ^2^
Plastic tube containing silica-coated film (Terumo Venoject II)	Appendix A	Appendix A
silica-coated plastic tube (Nipro Neotube)	Appendix A	Appendix A

^1^ Genuine A-PRF matrix prepared by an approved tube and a third-party’s centrifuge. ^2^ Genuine CGF matrix prepared by a conventional plain glass tube and an approved centrifuge.

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
