# Peer review of "Striking Differences in Platelet Distribution between Advanced-Platelet-Rich Fibrin and Concentrated Growth Factors: Effects of Silica-Containing Plastic Tubes"

_jfb, 2019, doi:10.3390/jfb10030043_

Round 1

Reviewer 1 Report

in PRF Matrix Preparation:  -the type of BD Vacutainer tubes used should be specified;

-cannot be defined as "A-PRF protocol" the blood product at a centrifugation of 200 x g for 14 minutes (the correct Choukroun protocol is 1300 g x 8 min); the centrifugation performed would be more like an i-PRF + performed for 14 minutes (700 rpm x 5 minutes); I would define it as "low-speed A-PRF protocol" "Low-speed A-PRF";

-it would be useful to have a processing temperature defined in °C (range);

Author Response

in PRF Matrix Preparation:  -the type of BD Vacutainer tubes used should be specified;

Response: We have added their product code (PRF366430) to the text.

-cannot be defined as "A-PRF protocol" the blood product at a centrifugation of 200 x g for 14 minutes (the correct Choukroun protocol is 1300 g x 8 min); the centrifugation performed would be more like an i-PRF + performed for 14 minutes (700 rpm x 5 minutes); I would define it as "low-speed A-PRF protocol" "Low-speed A-PRF";

Response: Ghanaati et al. [J Oral Implantol, 40(6):679-689;2014] prepared advanced PRF using sterile plain glass-based vacuum tubes (A-PRF10 tube) under the centrifugal conditions of 10 mL and 1500 rpm for 14 min. The same protocol was employed by Ehrenfest [Platelets, 29(2):171-184;2018].

Here, when this centrifugal speed (rpm) was converted to g-force in case of a Duo centrifuge, the centrifugal force was 264 × g. When your value (1300 rpm) is applied, it is calculated to be 198 × g, a value, which is almost identical to our centrifugal force (200 × g). Thus, we think that “the correct Choukroun’s protocol” you mentioned is the same as ours in terms of centrifugal speed (force). At least, “low-speed A-PRF protocol” would not be appropriate.

As for the duration of centrifugation, “8 min” is sometimes a little too short for samples with lower coagulation activity. In such a case, after centrifugation, the samples should be left for a brief time (~10 min). To avoid such sample-dependent variations, we fixed the duration at 14 min in accordance with the original PRF protocol. In this context, this protocol may be expressed as “short-term A-PRF protocol.”

To our knowledge, Choukroun has continuously modified his original protocol. Thus, it is difficult to identify which protocol is “correct”; what we can do is just update his concepts and protocols. Alternatively, we propose that the centrifugal conditions should be roughly classified into several combined categories (e.g, low, medium, high; 5 min, 10 min, 15 min, etc.). Thus, we can easily recognize which category our protocol can be classified into and group many modified protocols.

-it would be useful to have a processing temperature defined in °C (range);

Response: We prepared the samples at ambient temperature, which was 22-25℃. We have inserted this information into the Materials and Methods section.

Reviewer 2 Report

In this manuscript, the authors are studying the possible differences in platelet distribution in PRF matrices prepared by different protocols and using different tube types.

Major comments:

Results open to interpretation

The main issue with the manuscript is the lack of quantitative studies to support the conclusion of the authors. The results solely rely on immunohistochemical examination of the PRF and can, therefore, be easily be open to interpretations. For example, the authors claim that there are “striking differences in platelet distribution between the silica-coated plastic tubes and conventional plain glass tubes” but, based on only the observation of the immunohistochemical examination, beside for the PRFs obtain in the BD plain glass tube at high speed centrifugation and the one obtain in Terumo plastic tube at low speed centrifugation that have very distinctive patterns, all the other PRF are very similar.

As a result, the following conclusions can be questioned:

Line 217: We presently demonstrate that the same centrifugal conditions result in comparable distribution of platelets in CGF and A-PRF matrices, regardless of the brands of glass tubes used.

Line 219: we think that these findings are representative of all plain glass tubes approved by regulatory authorities

Line 229: the present data demonstrate that their assumption is wrong.

Line 262: We demonstrated that, as expected, low-speed centrifugation distributed platelets almost uniformly in the resulting PRF matrix.

The addition of quantitative studies would strengthen the authors’ conclusions and take away any doubt.

Conclusion not supported by results

Some results are missing from the manuscript and consequently some conclusions are not supported by data. For example:

Line 239: In this study, we observed that silica micro- or nano-particles were easily detached from the inner wall of the tube during blood collection and after retrieval of the PRF clot.

Minor comments:

Figure set and results part are using different name for the tubes, which makes it difficult to follow. For example, figure 3, “BD” in the figure, “Vacutainer” in the text (caption and result parts).

Line 151: “Other CD41+ platelets were distributed sparsely in the deep region and around the lower peripheral region (Figure 2B).” Not clear which other CD41+ platelet the authors are talking about.

Line 204: Capital “I” is missing at the beginning of the sentence.

Line 240: “Silica microparticles detached from inner wall of plastic tubes are thought to cause ubiquitous activation of both intrinsic coagulation pathway and platelets, which is consistent with the finding that platelets were distributed relatively more widely even after high-speed centrifugation.” References are missing

Author Response

Major comments:

-Results open to interpretation

-The main issue with the manuscript is the lack of quantitative studies to support the conclusion of the authors. The results solely rely on immunohistochemical examination of the PRF and can, therefore, be easily be open to interpretations. For example, the authors claim that there are “striking differences in platelet distribution between the silica-coated plastic tubes and conventional plain glass tubes” but, based on only the observation of the immunohistochemical examination, beside for the PRFs obtain in the BD plain glass tube at high speed centrifugation and the one obtain in Terumo plastic tube at low speed centrifugation that have very distinctive patterns, all the other PRF are very similar.

Response: Actually, we made the title more attractive after examining many samples and summarizing the findings. However, for reviewers who can observe only representative images presented in the manuscript, such a title may indeed seem overemphasized. However, we believe you can understand that this title is not overemphasized. To help readers understand this, we have further added a table (Table 2) collecting the representative images for easier comparison and have modified the corresponding results.

-As a result, the following conclusions can be questioned:

-Line 217: We presently demonstrate that the same centrifugal conditions result in comparable distribution of platelets in CGF and A-PRF matrices, regardless of the brands of glass tubes used.

Response: CGF and A-PRF matrices were prepared by the high- and low-speed centrifugation protocols, respectively. We have added this note to the sentence you indicated.

-Line 219: we think that these findings are representative of all plain glass tubes approved by regulatory authorities

Response: It is impossible to check all the glass tubes at once. However, as far as we have checked, the plain glass tubes, such as BD Vacutainer and A-PRF+, approved by regulatory authorities of individual states or unions, are highly and reproducibly capable of stimulating coagulation. This implies the high quality of these products. In contrast, some unapproved glass tubes imported for laboratory experiment only, did not stimulate coagulation. We have obtained evidence that this failure is due to silicon coating on the inner wall. However, because this is not the present focus of the study, we did not dwell further on this.

Here, what we want to claim is that the reproducibility of PRF preparation cannot be guaranteed using unapproved glass tubes.

-Line 229: the present data demonstrate that their assumption is wrong.

Response: In our previous study [Tsujino et al., Biomedicines, 2019], we demonstrated the contamination of tube-derived silica microparticles in the resulting PRF matrix. In this study, we demonstrated that even though centrifugation was done at a high speed, the existence of silica microparticles enabled platelets to aggregate and attach to fibrin fibers of the deep regions of the PRF matrix. Because the coagulation cascade is triggered by direct contact between silica microparticles and coagulation factor XII, this finding suggests that silica microparticles invade into the deep regions of the PRF matrix.

Recent in-vitro and in-vivo studies have provided evidence that amorphous silica are cytotoxic. Thus, silica microparticles that detach from the inner wall and embed into the deep regions of the PRF matrix may be released along with growth factors to act as a cytotoxic agent, as fibrin matrix degrades at the implantation sites.

In our parallel study [manuscript in submission], as expected, we clearly demonstrated the cytotoxicity of those silica microparticles in human primary cultures of periosteal cells.

Thus, we mentioned this sentence to alert clinicians who are not familiar with silica and believe silica-coated plastic tubes to be a reliable alternative of conventional plain glass tubes.

-Line 262: We demonstrated that, as expected, low-speed centrifugation distributed platelets almost uniformly in the resulting PRF matrix.

Response: In case of low-speed centrifugation, the platelets were distributed almost homogenously regardless of the tube types. As shown in Table 2, in contrast, high-speed centrifugation produced several minor differences among those tube types. In general, glass tubes were less effective in inducing homogenous platelet distribution than silica-containing plastic tubes. Therefore, we have modified the results in the text.

-The addition of quantitative studies would strengthen the authors’ conclusions and take away any doubt.

Response: We strongly agree with your opinion. However, to our knowledge, the current technology does not enable us to routinely and accurately determine platelet counts in certain parts of PRF matrices. It has been claimed that platelets can be counted in HE-stained specimens. However, this is incorrect; HE stains platelets light pink, which cannot be distinguished from the surrounding matrices or debris. Furthermore, because platelets are aggregated like a tightly-bound large cluster, they cannot be dissected on the image and counted individually upon activation on fibrin fibers.

Thus, we are aware that this immunocytochemical technique provides just semi-quantitative information. However, we believe the present findings, along with our previous and pending findings, are valuable enough to alert clinicians in terms of a potential hazard to patient health.

-Conclusion not supported by results

-Some results are missing from the manuscript and consequently some conclusions are not supported by data. For example:

-Line 239: In this study, we observed that silica micro- or nano-particles were easily detached from the inner wall of the tube during blood collection and after retrieval of the PRF clot.

Response: We have added our previous article as a reference to support this description [Tsujino et al., Biomedicines, 2019]. However, it is acknowledged that silica microparticles coated on the inner wall of tubes are easily and immediately detached by inclusion of liquid, such as blood and water. This explains why silica-coated plastic tubes facilitate coagulation that is more extensive and occurs faster than that occurring in conventional plain glass tubes.

Minor comments:

-Figure set and results part are using different name for the tubes, which makes it difficult to follow. For example, figure 3, “BD” in the figure, “Vacutainer” in the text (caption and result parts).

Response: To help readers understand, we added individual brand names in Table 1.

-Line 151: “Other CD41+ platelets were distributed sparsely in the deep region and around the lower peripheral region (Figure 2B).” Not clear which other CD41+ platelet the authors are talking about.

Response: We have added asterisks to indicate those platelet-poor regions in Table 2. Double-asterisks represent wider open spaces where platelets are sparsely distributed, while single-asterisks represent relatively smaller spaces.

-Line 204: Capital “I” is missing at the beginning of the sentence.

Response: We added “I” to give a correct word “It.”

-Line 240: “Silica microparticles detached from inner wall of plastic tubes are thought to cause ubiquitous activation of both intrinsic coagulation pathway and platelets, which is consistent with the finding that platelets were distributed relatively more widely even after high-speed centrifugation.” References are missing.

Response: We did not have clear evidence for the ubiquitous action of the coagulation pathway and platelets. However, considering the mechanism of coagulation, it is plausible that silica microparticles invade into deep regions and contact coagulation factor XII to induce platelet activation and aggregation on newly formed fibrin fibers. The present data of platelet distribution support this possibility. We have revised this passage with the inclusion of a reference.

Round 2

Reviewer 2 Report

The table 2 summarizes efficiently and clarifies the results. I would have it bigger as a figure and would put the other figures in supporting documents.

Author Response

-The table 2 summarizes efficiently and clarifies the results. I would have it bigger as a figure and would put the other figures in supporting documents.

Response: Thank you for your advice. We have enlarged individual images shown in Table 2 and combined them in new Figure 2. Other figures (previous Figs.2-5) are now presented as supplemental data (Figs. S1-S4).